# Environmental uncertainty shapes human effort learning

Rong Bi[1,2*], Jan Grohn[2,3], Patricia L. Lockwood[4,5], Miriam C. Klein-Flügge[1,2,3☉*], Lilian Weber[1,2,6☉*]

**1** Department of Psychiatry, University of Oxford, Warneford Hospital, Oxford, United Kingdom, **2** Oxford Centre for Integrative Neuroimaging (OxCIN), Centre for Functional MRI of the Brain (FMRIB), University of Oxford, Oxford, United Kingdom, **3** Department of Experimental Psychology, University of Oxford, Oxford, United Kingdom, **4** Centre for Human Brain Health, School of Psychology, University of Birmingham, Birmingham, United Kingdom, **5** Institute for Mental Health, School of Psychology, University of Birmingham, Birmingham, United Kingdom, **6** Institute for Cognitive Science, Osnabrück University, Osnabrück, Germany

☉ Shared senior authors.
* rong.bi@psych.ox.ac.uk (RB); miriam.klein-flugge@psy.ox.ac.uk (MCK-F); lilian.weber@uni-osnabrueck.de (LW)

## Abstract

Humans show remarkable flexibility in adapting their behaviour to constantly changing environments. This flexibility relies on the ability to regulate motivation in response to changing motivational demands. Typically, the amount of effort required to achieve a certain goal is not precisely signalled by the environment but needs to be learnt from experience. By contrast, prior work examining motivated choices has usually directly instructed effort requirements. It therefore remains unclear how healthy individuals estimate and flexibly regulate effort and how they might achieve this in dynamically changing environments. In the current study, we examine how effort learning is shaped by different types of environmental uncertainty when motivational requirements are not explicitly instructed. Analogous to tasks in the reward learning domain, we designed a novel effort learning task that systematically manipulated two key sources of uncertainty: volatility and noise. Participants were asked to exert effort by squeezing hand-held dynamometers. We characterised effort learning across different stages of the effort production process (e.g., initiation of effort production, effort expectation, error-driven adjustment), which allowed us to capture the dynamics underlying effort estimation and regulation over time. Our findings reveal that humans are able to learn effort requirements by integrating both effort priors and sensorimotor feedback. We further show that effort learning is modulated by environmental statistics, with slower force initiation, weaker priors, slower learning, and faster within-trial force adjustments in high noise environments, but slower learning and slower within-trial force adjustments in high volatility environments. In summary, when effort information is not instructed, different sources of uncertainty about an action's required effort are integrated into participants' internal priors to flexibly guide

**Data availability statement:** All raw data and individual and group-level summary data shown in the figures are available on OSF (https://doi.org/10.17605/osf.io/cf69x). Analysis code to reproduce the figures and results is available on GitHub (https://github.com/RongBi/Effort-learning-one-option), and on Zenodo (https://doi.org/10.5281/zenodo.19667752).

**Funding:** RB was supported by a scholarship from the China Scholarship Council (chinese-scholarshipcouncil.com). MCKF was funded by a Wellcome Trust/Royal Society Sir Henry Dale Fellowship (223263/Z/21/Z) (https://wellcome.org/), a UKRI EPSRC Frontiers Research Guarantee/ERC Starting Grant (EP/X021815/1) (https://www.ukri.org/) and a Leverhulme Award in Psychology (https://www.leverhulme.ac.uk/). PLL was supported by a Jacobs Foundation Research Fellowship (https://jacobsfoundation.org/), a Wellcome Trust/Royal Society Sir Henry Dale Fellowship (223264/Z/21/Z), a UKRI EPSRC Frontiers Research Guarantee/ERC Starting Grant (EP/X020215/1), and a Leverhulme Prize (PLP-2021-196). The study was also supported by the NIHR Oxford Health Biomedical Research Centre (NIHR203316) (https://oxfordhealthbrc.nihr.ac.uk/). The funders had no role in study design, data collection and analysis, decision to publish, or preparation of the manuscript.

**Competing interests:** The authors have declared that no competing interests exist.

**Abbreviations:** MVC, maximum voluntary contraction; PE, prediction error; RT, reaction time; SEM, standard error of the mean.

effort exertion. Our work may provide a useful framework for understanding motivational disorders where abnormal effort learning and estimation may underlie the reduced willingness to exert effort for reward.

## Introduction

An ancient proverb from the Chinese Philosopher Lao Zi states, "A journey of a thousand miles begins with a single step". This saying emphasises that achieving challenging goals requires initiating and consistently maintaining effort over time. In everyday life, the precise effort or motivation that is required to produce behaviour and achieve a goal is typically not instructed to the agent beforehand. Instead, effort requirements can be learnt through experience, for instance, via sensorimotor feedback from muscles or through repeated experience. By contrast, up until now, the classic approach for studying effort processing involves paradigms that directly instruct the required effort [1–10]. Moreover, upcoming motivational/effort requirements can be uncertain and can vary due to changes in the environment (e.g., the effort required for cycling to work is hard to predict perfectly and requires more adjustments in windy compared to calm weather conditions). However, little is known about how humans learn and estimate motivational/effort requirements in dynamically changing environments.

Effort learning, i.e., the process of forming and updating expectations about required motivational demands, represents an essential aspect of normal everyday life. For example, decreased physical activity has been linked to steeper effort discounting of rewards [11]. Such a reduced motivation to initiate and engage in activities in physically inactive individuals (e.g., sedentary individuals) has been speculated to originate in maladaptive effort learning or overestimation of the required effort [12]. In this way, aberrant effort learning processes may contribute to major public health challenges such as obesity. Similarly, motivational symptoms such as anhedonia and apathy have been linked to altered cost-benefit decision-making, including a reduced willingness to exert effort for reward [9,13–16]. Often in these studies, the required effort was signalled to participants. However, in real life, where effort requirements typically have to be learnt, these impairments could not only result from overestimating effort cost, underestimating future benefits, or failing to integrate cost-benefit information [17,18], but might actually be due to maladaptive effort learning. However, despite growing awareness of effort estimation biases in neurological and psychiatric disorders, it remains unclear whether and how healthy individuals can accurately estimate and flexibly regulate effort in changing and uncertain environments.

Predictive processing accounts would suggest that the brain forms internal models about upcoming efforts, generating predictions that can be updated when they deviate from reality. For building appropriate internal models for expected effort demands, it is important to know the uncertainty around one's estimate of required effort. For example, even when the effort associated with a goal has been experienced, there might be uncertainty around this estimate, and the same action may not be equally effortful next time (e.g., it takes more effort to cycle to work if sleep quality was poor

the night before). In the domain of reward learning, a distinction has been made between two sources of uncertainty: unexpected uncertainty ("unknown unknowns") refers to unpredictable and fundamental changes often studied as environmental volatility, while expected uncertainty ("known unknowns") refers to random noise that is stable and can be known [19–21]. While previous studies in the effort domain have shown that humans can learn effort requirements and minimise effort exertion when required efforts are not explicitly instructed [22–25], we do not currently know whether people use internal models of effort and whether effort learning is influenced by different sources of uncertainty in the environment.

The effect of different sources of uncertainty on learning has primarily been studied in the reward domain, where different types of uncertainty are predicted to have different effects on learning [26–32]. Specifically, higher volatility should increase the learning rate during reward learning because prediction errors are larger or more structured than expected. This means the weight assigned to new information should increase because the environment has changed and old information is outdated [19,33,34]. By contrast, higher noise in the form of predictable random changes should cause expected prediction errors and reduce the usefulness of new information and should therefore decrease the learning rate [32,34–37]. Recent work has further demonstrated that humans can simultaneously infer the volatility and noise during reward learning, and that the distinction between these sources of uncertainty is critical for understanding mental health symptoms [34,37]. Similarly, sensorimotor learning is susceptible to environmental volatility and sensory noise. For example, Herzfeld and colleagues [38] asked participants to make reaching movements under force-field perturbations and found that rapidly switching environments reduced sensitivity to error, resulting in slower learning. Using the same paradigm, Castro and colleagues [39] manipulated four levels of environmental consistency and found that motor adaptation was faster in more stable environments. Moreover, Körding and Wolpert [40] manipulated sensory uncertainty by adding Gaussian noise to visual feedback during a visuomotor displacement task. Their results revealed that humans integrate prior knowledge with sensory evidence, decreasing the influence of noisy sensory feedback during hand trajectory learning. Together, these findings illustrate that internal models of bodily control might also be flexibly updated depending on the statistics of the environment in which the agent is learning, which suggests effort learning may follow similar principles.

The aim of the current study was to examine how effort learning is shaped by different types of environmental uncertainty. Unlike reward learning (which focuses on external stimuli) or motor learning (which focuses on motor planning and execution), effort learning relates to the internal representation of a motivational cost and the regulation of motivation. We therefore designed a novel effort learning task that systematically manipulated the volatility and noise of motivational demands. Participants were asked to exert effort by squeezing hand-held dynamometers. However, no explicit information about the effort was provided and thus, effort requirements were not directly instructed. We analysed effort learning across different stages of the effort production process (e.g., initiation, effort expectation, adjustment), which allowed us to capture the dynamics underlying effort estimation and regulation over time. We found that participants formed priors (expectations) about required trial-wise effort levels, and that these priors were shaped by different sources of uncertainty. First, force initiation became slower and more hesitant in higher noise environments. Second, the formation and updating of effort priors was shaped by uncertainty, with weaker priors and slower learning in high compared to low noise, and slower learning in high compared to low volatility. Finally, people were faster to adjust their effort in higher noise environments. Our findings show that participants integrate different sources of uncertainty about required effort into internal priors, allowing flexible effort regulation in the absence of explicit effort information. This bears similarities with predictive processing accounts in reward and motor domains. It also has important implications for understanding disorders associated with apathy, where effort production is impacted, which may be due to maladaptive priors.

## Results

Thirty-five healthy participants performed a novel task designed to investigate participants' effort learning when required efforts were not instructed and changeable. In the task, participants were presented with a single option on each trial. They were instructed to produce the required effort as quickly as possible and hold it at the required level for three

seconds (Fig 1A). Efforts were produced by squeezing a hand-held dynamometer. Importantly, the required effort was not visually signalled: the effort target (lower bar of the yellow target zone in Fig 1A) was always shown at the same mid-level position of the "thermometer", its appearance simply indicated when participants could start to squeeze. Thus, in a low effort trial, the fluid in the thermometer would move to the central position easily when producing only a low level of force, while a high effort trial would require a strong squeeze for the fluid to reach the same mid-level position of the target zone. Participants completed four blocks of the task with each hand, for a total of eight blocks. Across blocks, the required effort varied in its expected and unexpected uncertainty. In other words, we manipulated the noise and volatility of required

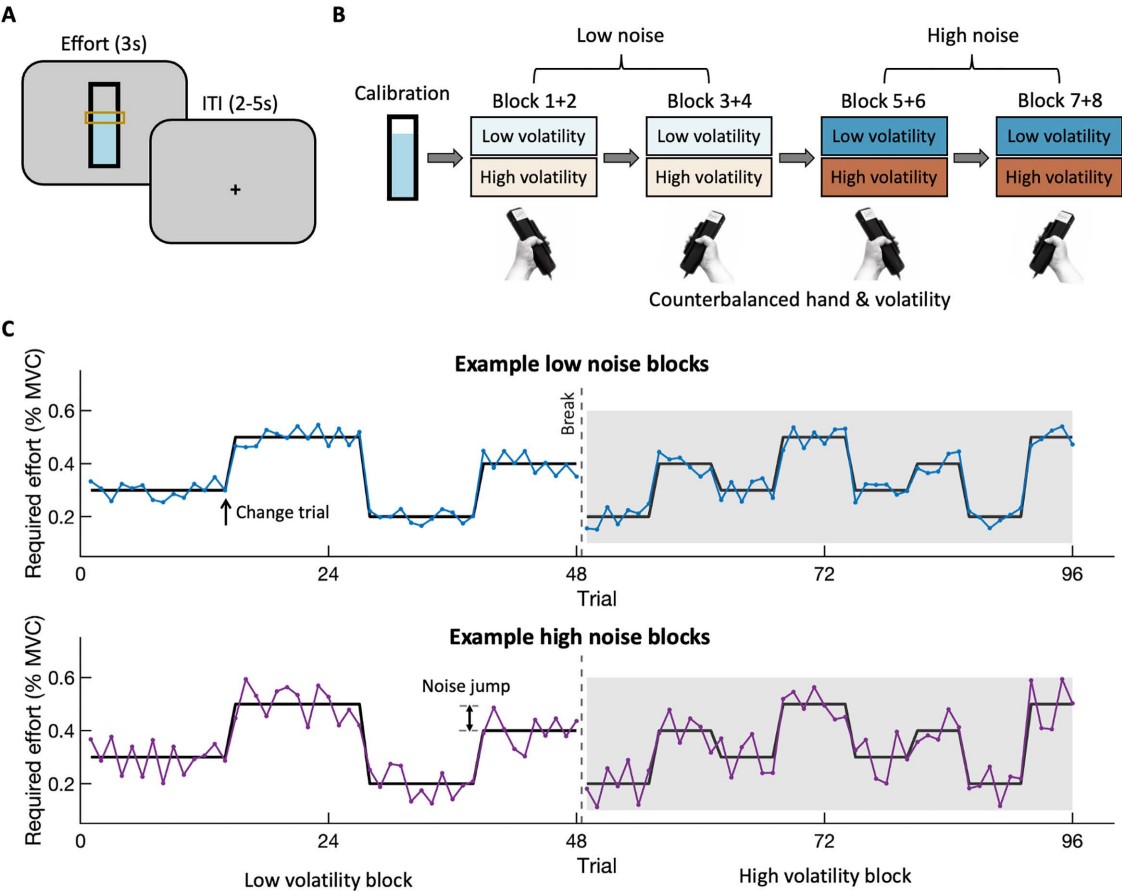

**Fig 1. Task and experimental procedures. (A)** An example trial in the effort learning task. Participants were instructed to squeeze the force dynamometer ("gripper") as quickly as possible to reach the effort target and maintain their grip force within the yellow target zone until the thermometer disappeared. The location and size of the target zone were fixed across trials, with the lower boundary appearing at the centre of the thermometer. Although its position was fixed, it indicated the level of required effort without giving explicit information about the required force. In other words, in a low effort trial, moving the fluid along to reach the target zone was easy and in a high effort trial, elevating the fluid to reach the target zone required a stronger squeeze. The required effort varied from trial to trial, and the duration of successful effort production (i.e., the length of time during which participants reached the lower boundary of the target zone while also maintaining the blue fluid within the indicated zone) determined task performance and payment. **(B)** Experimental procedure including calibration and eight task blocks. The force of both hands was calibrated by measuring the maximum voluntary contraction (MVC) for left/right hand separately prior to the task, which was set to 100% for each hand. Hand assignment and volatility blocks were counterbalanced across participants. The same hand was never used in more than two consecutive blocks. Photograph by the author. **(C)** Schedule of required effort for two example blocks. The white zone on the left shows an example of a low volatility block, and the grey zone on the right shows an example of a high volatility block. The black line is the average required effort, but actual effort requirements were noisy around that mean and are shown in two colours for low and high noise blocks (blue: low noise; purple: high noise). The single-headed arrow of the schedule shows an example of a change trial caused by volatility, and the double-headed arrow shows an example of a jump from the previous trial to the current trial caused by noise.

effort levels (Fig 1B). The average required effort changed in a step-wise fashion seven times in a high volatility block and three times in a low volatility block (Fig 1C, 'change trials') and noise around the mean was drawn from a wider uniform distribution in high noise compared to low noise blocks. In this way, we expected that this task might encourage participants to learn and anticipate the changing effort requirements over time.

## Effort execution shows evidence of learnt effort priors

We first examined the raw force-data (S1 Fig) to assess if people had formed priors, i.e., if we could find evidence for effort expectation and learning in our task. We focused on three distinct stages of the trial.

First, the initial latency between trial onset where the thermometer appears until participants start squeezing (force initiation). To identify the onset of squeezing, we computed the derivative of each force trace and detected the earliest time point with the greatest rate of change (see Materials and methods). The time from thermometer onset (i.e., trial onset) to the squeezing onset, termed start reaction time (start RT), was used as a measure of force initiation (Fig 2A).

Second, the force rapidly ramps up to a first "plateau" which may reflect participants' expectation of the required effort given any potential learning. To quantify effort expectation, we smoothed each force trace, computed the absolute derivative, and identified the first plateau as the time point with the smallest rate of change (see Materials and methods). The force at the first plateau (i.e., its height) was used as a measure of the participant's effort prior—their internal estimate of the required effort (Fig 2B).

Third, we quantified any adaptation from the initial expected effort level for effort fine-tuning until reaching a steady state (force adjustment). Once the effort prior was reached, the participant could visually assess its deviation from the actual target based on the level of the blue fluid and adjust accordingly. To quantify the duration of this adjustment stage, we computed the summed derivatives of the force trace within a sliding window to detect the steady state, defined as the smallest summed rate of change (see Materials and methods). The time from the effort prior to the onset of the steady state (termed adjustment RT), was used as a measure for the speed of abandoning and correcting the existing prior (Fig 2C). To examine whether start RT and adjustment RT were independent, we computed the correlation between log-transformed start RT and log-transformed adjustment RT across trials for each participant and then tested the Fisher-z-transformed correlation coefficients against zero using a one-sample $t$ test. We did not observe a significant effect (mean r = 0.016, $t(27)$ = 1.167, $P$ = 0.2533), suggesting that force initiation and force adjustment reflect independent learning-related processes.

Having developed measures that allowed us to quantify any potential learning, we next inspected if these did indeed reflect any learning of effort requirements. We included 28 participants in all main analyses following exclusion of seven participants who showed poor task engagement (see Materials and methods). The remaining 28 participants reached a success rate of 85%−100% (Fig 2D), meaning they reached the required effort level and maintained their force at the required effort level over 1 s in at least 85% of trials.

We first focused on our measure of **prior** to confirm that participants learnt and thus anticipated required effort levels without explicit visual information about the required force. We derived two ways to quantify effort expectations in our data. First, participants' priors, i.e., the force level at the initial plateau in a given trial $t$, closely followed their last experienced effort, i.e., the force required in trial $t-1$. Statistically, we verified this by plotting the difference between the previous trial's required effort and our measure for the current trial's effort prior for all trials and participants (Fig 2E). The mean of this distribution was centred narrowly around zero, suggesting that our measure of the learnt prior was precise at capturing participants' effort expectation.

The second indication for the presence of effort priors emerged when examining change trials. In change trials, participants tended to overshoot or undershoot at the initial plateau when a downwards or upwards change in required effort had occurred, respectively. This could be seen in the raw force traces (Fig 2B and 2C, see average and example force traces in S2A, S2B Fig) but also confirmed statistically. To do so, we computed the differences between the prior on trial $t$ and

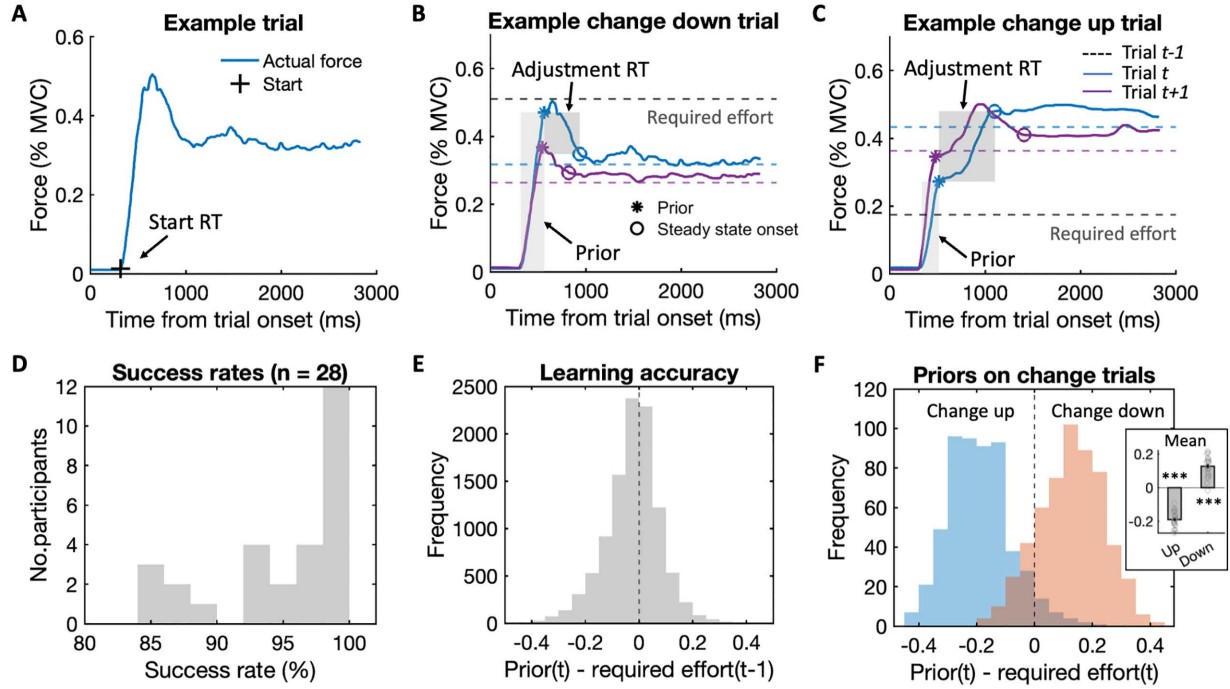

**Fig 2. Force traces show evidence of learnt effort priors. (A)** An example force trace illustrates the definition of the force initiation (plus sign). The start reaction time (RT) quantifies the latency of force initiation. **(B)** A typical force trace example from a downwards effort change trial. The blue solid line shows the force trace on change trial *t* and the purple solid line shows the force trace on trial *t+1*. The blue dashed line shows the new required effort level on trial *t* and the purple dashed line shows the subsequent trial's effort level, while the black dashed line shows the required effort level before the downwards change on trial *t-1*. The force trace shows that the overshoot between the first plateau (star) and the required effort (dashed line) decreases both within the trial and across trials. The force at the first plateau (height of the light grey shading) represents the effort prior—the participant's expectation of the required effort before any information from the current trial has been integrated into their learning. The duration (height of the dark grey shading) from the effort prior to the onset of the steady state (circle), termed adjustment RT, quantifies the speed of correcting the force within a trial. **(C)** An example upwards effort change trial and subsequent trial show the flipside, namely that there is an initial undershoot which reflects the effort expectation from before the effort level unexpectedly increased. **(D)** The distribution of success rates across the 28 included participants. A trial is counted as successful if the participant maintained the required effort for more than 1,000 ms, and the histogram shows the percentage of trials for all participants. **(E)** Priors on trial *t* on average accurately reflected the expected effort level as inferred from the required effort on trial *t-1*. The distribution across all trials in all participants shows the mean is centred narrowly around zero, indicating the learnt priors are accurate in capturing effort expectations. **(F)** *Left*: The difference between the current effort prior and current required effort in upwards change (blue) and downwards change (orange) trials across all participants. *Right*: The mean of this difference in each participant for change up and change down trials. The mean of both distributions is significantly different from zero, showing evidence that the prior on trial *t* was not confounded by information gathered on the current trial.

the effort requirement on trial *t* separately for upwards and downwards change trials (Fig 2F, left). Both distributions were significantly different from zero (change up: $t(27) = -21.617$, $P<0.0001$; change down: $(t(27) = 11.711$, $P<0.0001$; Fig 2F, right). Specifically, participants' priors were too low (undershoot) when the required effort changed unexpectedly from low to high (blue distribution), but their priors were too high (overshoot) when the required effort changed from high to low (orange distribution). The effort prior was defined at the initial plateau, which raises the possibility that sensorimotor feedback may have been integrated into the estimated prior. However, control analyses revealed only a minimal influence of online feedback at this early stage (prior duration: M = 230.32 ms, SD = 25.11 ms; S3 Fig). Thus, participants showed clear expectations for effort requirements and our measure of effort prior indeed captured the effort expectations they formed based on previous trials rather than effort requirements inferred or observed on the current trial.

PLOS Biology

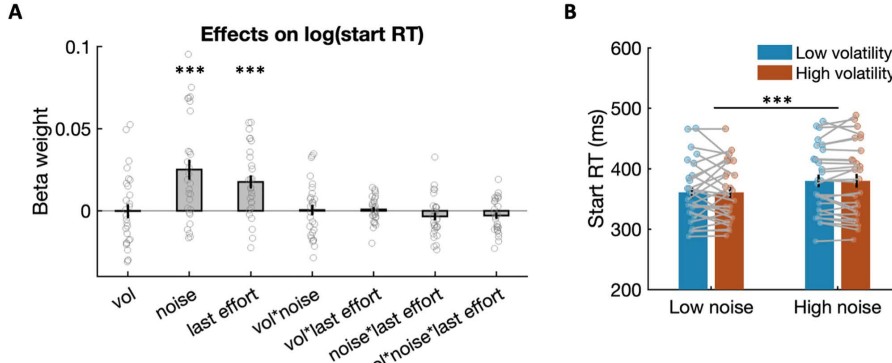

**Fig 3. Force initiation is slower under higher expected uncertainty. (A)** Group mean estimates derived from a linear regression shows the effects of noise and last effort (previous trial's required effort) on log-transformed start RT. The force initiation is slower in higher noise environments or after exerting higher effort. Stars indicate significance from a two-tailed one-sample *t* test on the obtained parameter estimates. **(B)** Raw start RTs in high and low volatility environments across low and high noise environments illustrate the slower RTs seen in high noise blocks. Error bars indicate the standard error of the mean (SEM). *** *P* < 0.001. All data used to generate this plot can be found in S1 Data.

## People's force initiation is slowed by higher expected uncertainty

Having established that people form priors and learn expectations about upcoming effort levels, we next asked whether environmental statistics might dynamically shape effort learning. To look at this, we considered all three stages in the force traces introduced above, namely force initiation (start RT), the initial plateau (prior), and force adjustment (adjustment RT).

We first asked whether the speed of effort initiation at the start of the trial, which may reflect the confidence and thus the strength of the formed prior, may change as a function of expected and unexpected uncertainty (noise or volatility). To address this, we estimated a linear regression model to identify factors influencing log-transformed **start RT** (see Materials and methods and Fig 3A). The results show that start RTs were significantly affected by noise, with more delayed starts or longer start RTs in high noise environments (*t*(27) = 4.130, *P* = 0.0003), indicating slower force initiation under greater expected uncertainty. We did not find an effect of unexpected uncertainty (volatility) on start RTs. Fig 3B illustrates raw start RTs across the four environmental conditions, showing slower start RTs in high compared with low noise environments. As higher noise increases the predictable uncertainty associated with the required effort, this delay may reflect reduced confidence in the learnt prior when initiating effortful actions.

Start RTs were also longer after experiencing higher effort requirements (*t*(27) = 4.396, *P* = 0.0002), possibly reflecting short-term fatigue from recent effort exertion. The low noise blocks were always presented before the high noise blocks, which raises the possibility that long-term fatigue might be a confound. However, control analyses revealed no gradual increase in average start RTs or the slope of start RTs over trials within blocks, suggesting that the observed noise effects are unlikely to be confounded by long-term fatigue (S4 Fig).

## Effort prediction and learning are shaped by environmental uncertainty

We next turned to the question of whether participants' **priors**, i.e., the initial phase of force production, where the first plateau in the force ramp expresses participants' force expectations, may be affected by the statistics of the environment. Force traces suggested that the prior can be estimated with good accuracy from the most recent effort (Fig 2E). An initial inspection of the prior's accuracy (i.e., the difference between the prior and the previous trial's experienced effort), separately for high and low noise environments (Fig 4A, left), indicated that the two distributions did not significantly differ in their means (*t*(27) = 0.853, *P* = 0.4010), were both centred around zero, but that their variance was significantly different: priors were more precise in low noise compared to high noise environments (*t*(27) = −3.936, *P* = 0.0005; Fig 4A, right).

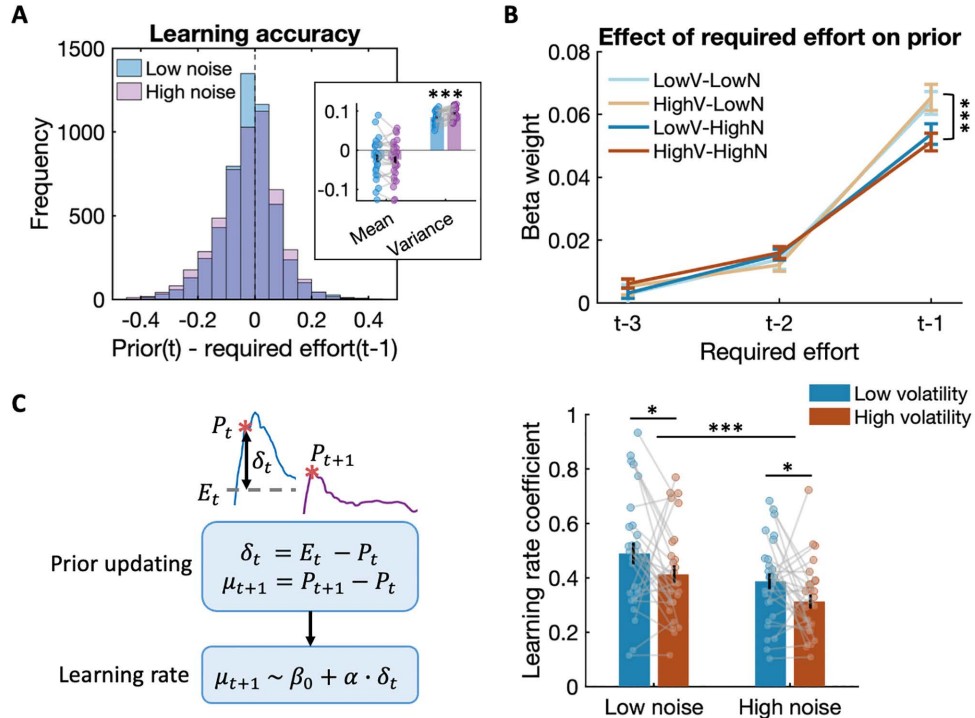

**Fig 4. Prior formation and updating are differentially affected by noise and volatility. (A)** *Left*: The two distributions show the learnt priors' accuracy by comparing the current trial *t*'s effort prior with the previous trial's required effort in low noise (blue) and high noise (purple) blocks, across all trials in all participants. *Right*: The mean and variance of learning accuracy (current prior – previous required effort) in each participant across low and high noise environments. The significant difference in variance indicates that the learnt priors are more accurate in low noise compared to high noise environments. **(B)** Required efforts experienced in the previous three trials inform the current trial's effort prior, and the required effort in trial *t-1* has less weight on the formation of effort priors in high noise compared to low noise environments. **(C)** *Left*: Schematic illustration and equations describing prior updating and the computation of learning rates. $P_t$: effort prior in trial *t*; $E_t$: required effort in trial *t*; $P_{t+1}$: effort prior in trial *t + 1*; $E_{t+1}$: required effort in trial *t + 1*; $\delta_t$: prediction error (PE) in trial *t*; $\alpha$: learning rate; $\beta_0$: intercept, representing the baseline value of updating when PE is zero; $\mu_{t+1}$: effort updating in trial *t + 1*. *Right*: Average learning rate in high and low volatility across high and low noise environments. This shows that a prediction error (PE) based learning model could capture the influence of environment on effort updating, with significantly slower learning in high noise compared to low noise environments and with slower learning in high compared to low volatility environments. Error bars indicate the standard error of the mean (SEM). *** *P<0.001. All data used to generate this plot can be found in S1 Data.

However, since average effort requirements remained relatively stable between change trials, priors might also form over longer timescales going back further than just one trial, and this formation might be influenced by the environmental statistics.

To test this, we fit separate linear regression models to each of our four different types of environments, modelling the current prior as a function of the required effort on the previous three trials (see Materials and methods and Fig 4B). We excluded change trials where effort levels changed more dramatically because we were interested in the influence of uncertainty on behaviour during phases with a constant mean effort. Across environments, past effort significantly influenced the current prior (trial *t-3*, *t*(27) = 3.294, *P*=0.0028; trial *t-2*, *t*(27) = 12.387, *P*<0.0001; trial *t-1*, *t*(27) = 20.502, *P*<0.0001), showing that efforts experienced more than one trial back still influenced current effort expectations.

Three-way ANOVAs (noise × volatility × trial) on the beta weights revealed significant main effects of trial (*F*(1,27) = 220.74, *P*<0.0001) and environmental noise (*F*(1,27) = 12.345, *P*=0.0015), with more recent trials and lower noise environments showing larger weights. Importantly, there was a significant interaction between trial and environmental noise (*F*(1,27) = 10.28, *P*=0.0002). The Bonferroni-corrected post hoc tests revealed a significant noise effect only for trial *t-1*,

with effort experienced on trial *t-1* having a weaker influence on priors on trial *t* in high noise environments compared to low noise environments ($P < 0.0001$). This suggests that effort priors are formed based on previous experience, but the most recent prior experience is down-weighted according to its reliability, with lower reliability and thus lower weights during periods of greater expected uncertainty.

Next, we wanted to know whether the statistics of the environment influenced how people update their effort priors across trials. We estimated the average learning rate in each condition using a model-agnostic approach [34,41]. As illustrated in Fig 4C (left), the PE was defined as the difference between the required effort on the current trial and the prior for that trial. **Prior updating** was quantified as the difference between the prior force on the subsequent trial and the current prior. As the learning rate was formally equal to the ratio of the prior update to the PE, we applied linear regression to estimate an average learning rate coefficient by regressing the update against the PEs across trials. This allowed us to quantify the main effects of both volatility and noise factors as well as their interactions. We excluded change trials and one subsequent trial, where effort levels changed suddenly, to avoid the estimates being driven by these large changes.

To test for learning differences between the environments, we applied a two-way ANOVA to the learning rate coefficients (Fig 4C, right). We observed significant main effects of the environment's volatility ($F(1,27) = 5.404$, $P = 0.0279$) and noise ($F(1,27) = 14.445$, $P = 0.0007$), both reducing the learning rate. There was no significant interaction between volatility and noise ($F(1,27) = 0.001$, $P = 0.9700$). This suggests that both forms of uncertainty weaken across-trial effort updates. To examine whether effort updates to volatility and noise were related, we computed the correlation between individual changes in log-transformed learning rate from low to high volatility and the corresponding changes from low to high noise. We did not observe a significant correlation between these learning rate changes ($r(27) = -0.179$, $P = 0.3625$), suggesting that sensitivity to volatility and noise reflects independent aspects of effort learning. We also examined whether learning is influenced by motor noise, but our control analyses found no evidence for this in our task (S5 Fig).

### Within-trial effort flexibility is higher under expected and lower under unexpected uncertainty

In our final set of analyses, we investigated participants' flexibility at adjusting their force levels *within-trial*, i.e., during the time after they had reached the effort prior and were experiencing a PE, defined as the distance between the reached prior (and thus indicated fluid level on the thermometer) and the required effort force on that trial (as shown by the target zone). We wondered whether environmental statistics might also affect how quickly people were able to correct their prior-based force level to reach the required force level. The force trace results (Fig 1A–1C) revealed a steady state closely approaching the required effort, suggesting that the duration between prior and steady state (termed **adjustment RT**, see above) could serve as an indicator of within-trial effort flexibility.

To identify factors influencing log-transformed adjustment RT, we applied a linear regression model (see Materials and methods and Fig 5A). The results revealed that adjustment RTs were significantly longer in high volatility compared to low volatility environments ($t(27) = 3.513$, $P = 0.0016$), indicating slower adjustments and thus a reduced flexibility under higher unexpected uncertainty. Importantly, this volatility effect was not driven by change trials, as it remained when we repeated the regression without change trials (S6A, S6B Fig). Conversely, adjustment RTs were significantly shorter in high noise environments ($t(27) = -3.567$, $P = 0.0014$), indicating increased flexibility potentially due to looser priors that could be abandoned more easily under higher expected uncertainty. Additionally, there was a significant interaction between volatility and noise ($t(27) = -2.791$, $P = 0.0095$). This interaction is visualised in Fig 5B, showing that the effect of volatility on adjustment RTs was larger during low noise, and the effect of noise on adjustment RTs was stronger during high volatility. To examine whether within-trial adjustments to volatility and noise were related, we computed the correlation between individual changes in log-transformed adjustment RTs from low to high volatility and the corresponding changes from low to high noise. We did not observe a significant correlation between adjustment RT changes ($r(27) = -0.098$, $P = 0.6177$), suggesting that sensitivity to volatility and noise reflects independent aspects of within-trial force adjustment. Adjustment RTs were also longer when the required effort increased from a low to a high level (jumpSign, $t(27) = 10.048$, $P < 0.0001$)

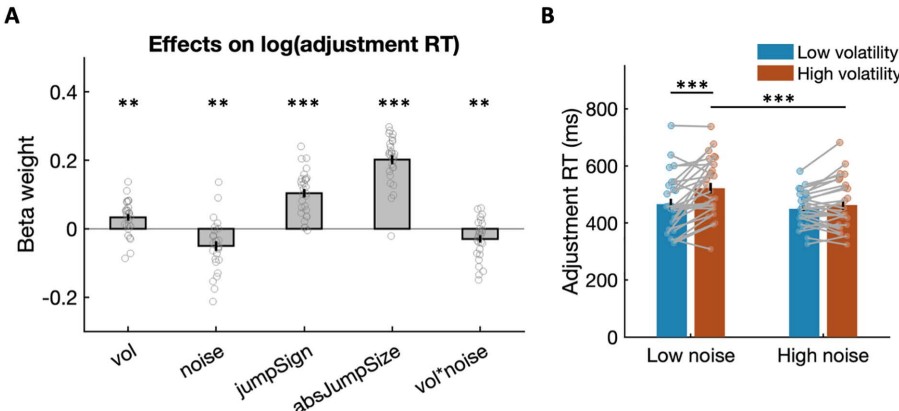

**Fig 5. Within-trial force adjustments reflect both noise and volatility. (A)** Group mean estimates from a regression explaining log-transformed adjustment RTs from various task factors. Stars indicate statistical significance in a two-tailed one-sample $t$ test testing the effect of model parameters on adjustment RTs. Force adjustments were faster in higher noise but slower in higher volatility environments. Also, trial-by-trial changes in effort—the direction and size of effort jumps both slowed down within-trial force adjustments. **(B)** Raw adjustment RTs in high and low volatility as well as high and low noise environments. This shows the interaction between noise and volatility, with slower force adjustments in high compared to low volatility in low noise environments, but faster adjustments in high compared to low noise in high volatility environments. Error bars indicate the standard error of the mean (SEM). * $P < 0.05$, ** $P < 0.01$, *** $P < 0.001$. All data used to generate this plot can be found in S1 Data.

or when the absolute magnitude of the change was larger (absJumpSize, $t(27) = 15.251$, $P < 0.0001$; Fig 5A). These findings highlight the sensitivity of within-trial force adjustments in response to both direction and size of trial-by-trial changes in effort requirements.

Compared to high volatility blocks, low volatility blocks in our task afforded higher precision in estimating the true (average) required effort, since participants had more trials to learn before the next change-point occurred. We wondered whether the increase in precision would explain why people were faster to correct their force to the required force level during low volatility blocks. To test this, we separately averaged adjustment RTs for early and late stages within each constant mean effort phase and found faster adjustments in late but not early stages when comparing low with high volatility blocks (S6C, S6D Fig). Together, these findings suggest longer exposure and thus more experience in a relatively stable and certain environment allows participants to adjust their force faster to the true demands of the current trial.

## Discussion

To effectively achieve goals in uncertain and changing environments, it is crucial to anticipate required motivational demands and to flexibly regulate motivation and actions in response to potential changes. Volatility and noise are two sources of uncertainty, which prior work has shown to play important roles in reward learning processes. However, little is known about whether the ability to adapt to environmental statistics also applies to the effort domain and thus, whether humans can learn and flexibly adjust efforts to dynamically changing environments. To address this question, we conducted a behavioural experiment to investigate how human participants learn effort in volatile and noisy environments. Our findings revealed that humans form expectations and use effort priors as well as real-time sensorimotor feedback to inform their effort exertion when explicit information about effort requirements is lacking. Moreover, effort learning is shaped by environmental statistics. Specifically, force initiation and prior updating across trials are slowed, but within-trial force adjustments are accelerated under higher expected uncertainty (noise). By contrast, across-trial prior updating as well as within-trial force adjustments are slowed by higher unexpected uncertainty (volatility).

We defined a way to measure the expected effort given the participants' initial force plateau. Notably, we found that these effort priors capture true effort expectations, rather than current effort requirements that could be inferred on-the-fly.

While previous studies have demonstrated humans' ability to learn effort from previous experience [22–25], the dynamic processes of effort learning within and across trials have largely been overlooked. In the current study, we identified three stages of effort production that we believe capture the nature of the predictive processing that underlies effort learning: force initiation, ramp-up to prior, and prior adjustment. The force initiation stage reflects the individual's initial response to launch an effortful action, where the brain generates a motor command based on its prior belief about the required effort. During the ramp-up stage, force rapidly increases towards the learned effort prior, reflecting how internal priors guide the ongoing motor execution. Although we cannot entirely exclude the possibility that sensorimotor feedback contributed to the estimated prior, our control analyses suggest that such influence was minimal. Finally, the prior adjustment stage captures the integration of real-time sensorimotor feedback to correct for errors between the learned prior and the actual required effort, reflecting error-driven adaptations. This experienced error is then used to update the prior effort belief for the next trial. Taken together, our findings suggest that the brain infers the required effort and regulates effort exertion in real-time by integrating priors with sensory evidence, preventing overfitting that leads to unnecessary exhaustion, and underfitting that would fail to meet effort demands, consistent with Bayesian principles of predictive processing.

In our data, the first key analysis looking at the influence of environmental statistics highlighted effects on the speed of force initiation. We found that the force initiation was slowed down in higher noise environments. Consistent with this finding, Izawa and Shadmehr [42] reported longer reaction times to start a reaching movement when the uncertainty about the target location increased. In their study, uncertainty was manipulated by varying the standard deviation of Gaussian blobs indicating the target location, analogous to the noise added to the average required effort in our effort task. By contrast, in tasks where effort was explicitly signalled and therefore certain, force initiation was generally faster for higher effort levels [43]. Combing previous studies with our own findings, we suggest that as expected uncertainty increases, people may become less confident in their learnt prior, resulting in a delay in generating a motor command. Notably, only noise but not volatility modulated the force initiation, consistent with the idea that volatility supports hidden state inference and that humans are able to distinguish the source of variation in effort learning—whether it reflects true change or random noise. Previous work has demonstrated the existence of short-term fatigue and long-term fatigue in effortful decision-making [44]. One limitation of our study was that we did not counterbalance the order of noise blocks, which made it difficult to completely dissociate noise effects from long-term fatigue. However, several control analyses consistently suggest that noise effects cannot fully be explained by long-term fatigue. Part of the accumulated fatigue was probably mitigated by breaks and by alternating hands between blocks. Furthermore, we revealed that the force initiation was slower when the last trial's effort was higher, reflecting short-term fatigue in trials following a high exertion of effort. Together, these findings suggest that the brain's motor planning involves an inferential process, where it estimates the reliability of its predictions by considering environmental statistics and integrates these with the body's current physiological state to optimise the timing of motor commands.

Our second key analysis examined the influence of environmental statistics on the formation and updating of effort priors. We found that effort priors were predicted by the required effort of the preceding trials, suggesting that the predicted effort relied on a longer history of effort experience going back at least three trials. Notably, the weight of the most recent effort was largest (compared to earlier trials) but reduced under higher noise compared to low noise. This suggests that increased expected uncertainty changes the relative importance of information received at different time points and promotes integration over longer timescales. Importantly, when we looked at changes in the effort prior across trials, we demonstrated that a PE-based learning model was able to capture this influence of environmental statistics on effort prior updating across trials. Specifically, we found that for effort learning, the learning rate was reduced under higher noise and higher volatility. Although the regression-based analysis and the PE-based model both characterise updating processes, they are not mathematically identical. The regression model captures the influences of a limited recent history (i.e., the past three trials with independent weights), whereas the RL model evaluates updating over the full effort history (with a fixed kernel shape). Differences in the observed volatility effect across the two approaches likely reflect differences in

temporal integration. While we focused on this standard PE-based model to characterise basic updating processes, other alternative models (e.g., asymmetric learning rates that depend on the sign of prediction errors) may provide additional insights and could be explored in future work. Based on our model, increased noise induces moment-to-moment fluctuations, which are thought to reduce the reliability of recent information, thus extending the window of integration. Similarly, increased volatility reduced learning rates in our study. This pattern is opposite to the increased learning in the reward domain [19,29,34], suggesting that effort learning may rely on different mechanisms for adapting to state uncertainty.

In our final analysis, we examined the influence of environmental statistics on the dynamic within-trial adjustment of force levels. We showed that effort priors were abandoned faster on-the-fly under higher noise environments, but slower when the volatility was higher. In other words, as expected uncertainty increased, priors were weaker and could be corrected more quickly. Consistent with this finding, Izawa and Shadmehr (2008) reported faster movement corrections after a target jump when feedback uncertainty was increased, because it resulted in greater reliance on the delayed sensory observations. However, increased unexpected uncertainty (volatility) led to slower corrections of existing priors. Two explanations may account for this finding. On the one hand, the slower adaptation within a trial, combined with the slower trial-by-trial prior updates, provides consistent evidence that in the motor/motivational domain, experiencing more frequent state changes makes the body slower to adapt to prediction errors, possibly because frequent changes encourage behaviours that conserve energy. On the other hand, faster corrections within-trial only occurred in trials that happened late, not early, during a constant mean effort phase, suggesting that the volatility difference was driven by the longer exposure to a constant mean in stable blocks. This longer exposure provides more experience to form precise predictions about the upcoming required effort, facilitating force adjustments. Indeed, previous theoretical and empirical work has pointed to opposing effects of volatility on learning: increases in learning rates due to heightened unexpected uncertainty on the one hand, and reductions in learning rate due to a decrease in the precision of predictions on the other hand [27,45]. In our data, the latter effect seems to dominate, as supported by both the trial-by-trial prior updates as well as the within-trial force adjustments. Apart from volatility and noise effects, within-trial force adjustments were also slower when the required effort increased and when the change from the previous trial to the current trial was larger. Together, these findings suggest that within-trial adaptations are sensitive to both environmental changes and recent changes.

The findings from this study add to a growing body of literature, primarily in the reward domain, suggesting that humans can adapt their learning to environmental statistics, and distinguish between different types of uncertainty. In terms of noise, previous studies generally reported reductions in reward learning rates with higher noise and reductions in motor learning with higher noise. Across domains, experiencing small, random, and predictable fluctuations should not prompt the brain to update the value and/or change the behaviour. This is consistent with what we find here in what we believe is the first study investigating the predictive processes underlying effort learning. By contrast, unlike noise, volatility exhibits opposite effects across domains: it increases reward learning rates [19,34] but decreases motor learning rates [38,39]. Our work is in line with studies looking at motor learning, providing support to the idea that unexpected uncertainty may have differential effects on learning in reward versus motor domains. We note that there may be important differences that could be causing this discrepancy. For example, in reward learning studies, volatility is manipulated as the rate of probability switches or the variance of the Gaussian diffusion noise in reward outcomes. Yet, in motor learning, consistent with our work presented here, volatility affects the consistency or demand related to motor execution and thus also changes the state of the body (e.g., muscle fatigue, recover frequency, disease) [46]. From this perspective, reward learning requires flexible inference about an external rule or hidden state, where the reward PEs reflect the rules of reward distribution, and thus fast adaptations to unexpected changes are helpful. By contrast, sensorimotor learning aims to refine internal motor models and maintain energy homeostasis, whereby the brain assumes that motor PEs reflect its own mistakes, thus over-adjusting to new contingencies is effortful and aversive. Therefore, our study suggests that effort learning of physical demands is more similar to motor learning, not reward learning. This has important implications for understanding how the brain interprets environmental statistics and adapts behaviours across domains.

 

We believe our work provides valuable insights for studying the mechanisms underlying effort learning in people with compromised motivation (e.g., anhedonia or apathy), which is common across neurological and mental health disorders. First, healthy participants can form effort priors and integrate these priors with sensorimotor feedback to regulate effort behaviours. This opens up new possibilities to study whether altered cost-benefit decision-making reported in depression, apathy, or in physically inactive individuals may result from impaired effort learning or effort overestimation. While previous frameworks have raised this possibility [17,18], our findings provide direct evidence of the process underlying dynamical effort estimation and regulation in a healthy population. Second, we show that effort learning is shaped by environmental statistics, which suggests a potential relationship between altered sensitivity to uncertainty (e.g., noise/stochasticity or volatility) and pathological motivation-related decision-making, which should be further explored. For example, individuals with mood and/or anxiety symptoms have sometimes been found to have difficulty in optimally using environmental statistics to adapt their decision-making [47–51]. Piray and Daw (2024) proposed that when the brain is less accurate in estimating one type of uncertainty (e.g., stochasticity), this can lead to compensatory overestimation of another type of uncertainty (e.g., volatility). Taken together, our study suggests several potential lines of work in motivational disorders that could aim to improve the accurate characterisation of effort impairments by probing effort learning and adaptation mechanisms.

The design of this study also leaves several open questions for future research. First, because reward learning was not directly manipulated, conclusions about cross-domain differences between effort and reward learning remain tentative. Second, we did not mention concepts related to environmental changes (i.e., volatility or noise) in the task instructions, nor did we assess participants' conscious awareness of their adjustments to these environmental statistics, making it difficult to determine whether these adaptations occurred explicitly or implicitly. Finally, we did not collect subjective reports of effort exertion. Previous work has shown that greater motor variability in exertion is associated with increased perceived effort [52–55], and future studies could examine how motor variability contributes to the subjective experience of effort and thereby influences motivational regulation.

The primary aim of this study was to investigate the influence of uncertainty on effort learning. However, effort expectation is a hidden variable that has not been directly quantified in previous work. By identifying several distinct and meaningful stages during effort production, we were able to capture effort priors and demonstrated effort prior adaptations both within and across trials. Interestingly, environmental statistics exhibited similar effects on effort when compared to motor learning, but partly opposite effects compared to their role in reward learning.

## Materials and methods

### Participants

Thirty-five right-handed young participants (25 females; age range 18–39, M = 26.17, SD = 5.73) with normal or corrected-to-normal vision were recruited from the Oxford Psychology Research participant recruitment scheme. Participants who had an injured wrist were excluded because the study involved physical force exertion with both hands. Before the experiment, all participants gave written informed consent. The study was approved by the Medical Sciences Interdivisional Research Ethics Committee (MS-IDREC-R84862/RE001) and conducted in accordance with the Declaration of Helsinki. Participants were remunerated with £12 for taking part in the study, plus a possible bonus of up to £3 depending on their task performance (range 13–15; M = 14.14, SD = 0.43).

### Experimental task

We developed a novel effort learning task where one option varied in its effort requirements on a trial-by-trial basis. In this task, the required effort level was never explicitly signalled to participants, thus requiring them to track and infer required effort levels over time.

As shown in Fig 1A, the trial began when the display showed a thermometer, indicating to participants that they were now required to exert effort to reach the yellow target as quickly as possible by squeezing a hand-held dynamometer

(TSD121B-MRI; BIOPAC MP160, USA). On the thermometer display, the level of required effort that participants had to achieve always corresponded to the centre of the thermometer, so the display did not provide any information about the required effort. However, real-time feedback on the exerted force was provided to the participants in the form of blue 'fluid' that moved up and down within the thermometer as they were squeezing with different forces. The gain or sensitivity of the fluid's response to the participant's force exertion, therefore, scaled with the required force on that trial. For instance, at higher effort levels, a greater force was required to elevate the fluid to the same target height compared to lower effort levels, where the fluid moved more easily. To ensure that the participants regulated their force exertion without exhausting themselves, the target was presented as a zone rather than a line. The height of the target zone corresponded to 10% of the height of the thermometer or 20% of the required effort in each trial. The thermometer was shown for 3 s, and participants were encouraged to keep the force within the indicated range until the thermometer disappeared. In other words, the participant's goal was to ensure that the force exerted in each trial stayed within the range of 100% to 120% of the required effort as long as possible. The longer participants succeeded at keeping the blue fluid within the target range, the more reward they would receive at the end of the experiment (bonus between £0 and £3 scaled directly with the time spent within the 100%–120% target zone). After effort exertion, no feedback was provided, but there was an ITI (i.e., fixation cross) lasting 2–5 s before the next trial started. The ITIs were randomised and balanced so that each duration (2, 3, 4, and 5 s) occurred once in each set of four consecutive trials, in a random order.

## Procedure

The study employed a 2 x 2 within-subject experimental design (Fig 1B) with factors volatility (low/high) and noise (low/high). To ensure sufficient trials, participants were instructed to complete all four conditions separately with each hand, making a total of eight blocks. At the beginning of the experiment, participant's grip strength was individually calibrated to determine their maximum voluntary contraction (MVC) for both hands separately. The individual MVC was then set to 100% of the force level for the remainder of the experiment, so that the levels of effort used throughout the study were achievable and comparable across participants.

Participants completed the eight task blocks in groups of two, such that each group contained one block of low volatility and one block of high volatility (counterbalanced; Fig 1B). There were four of these groups: Block 1 + 2, Block 3 + 4, Block 5 + 6 and Block 7 + 8. Two groups of blocks were completed with the right, and two with the left hand (start hand counterbalanced, interleaved to avoid exhaustion). For each hand, the two blocks with low noise were completed before the two blocks with high noise. Each block comprised 48 trials (5.5 min), and thus 96 trials were completed with one hand before switching to the other hand (total: 11 min).

In the low volatility block (Fig 1C), participants completed 48 trials in which the average required effort changed every 10−14 trials. By contrast, in the high volatility blocks, which contained the same number of trials overall, the average required effort changed every 5−7 trials. In other words, there were three changes in required effort during low volatility blocks and seven changes during high volatility blocks. The average required effort could take one of four levels: 0.2, 0.3, 0.4, and 0.5 of the participant's MVC. Actual required efforts were generated by adding noise to the average effort. Low noise was sampled from a uniform distribution ranging from −0.05 to 0.05 (mean = 0), while high noise was sampled from a uniform distribution ranging from −0.1 to 0.1 (mean = 0). The number of negative noise and positive noise values was balanced within each constant mean effort phase. Thus, the actual required effort ranged from 0.1 to 0.6 of MVC (Fig 1C).

## Statistical analysis

**Preprocessing.** The raw data were force traces recorded during participants' effort exertion in each trial. The sampling frequency of our gripper was 500 Hz, which means one data point (i.e., sample) was recorded every 2 ms. Trials from both hands were combined across all analyses. Force traces were visually inspected to ensure participants had followed task instructions and squeezed at the required levels. During each 3-s effort production, each trial was supposed to record

1,500 samples. Due to recording issues or brief squeezes, some trials had less than 1,300 samples and were excluded. This only affected eight trials across all participants. In five trials across all participants, the force trace began with nonzero values (mean of the first 20 samples > 0.2 MVC) due to accidental squeezing before the thermometer onset. In all of these trials, the force returned to zero by ~160 samples and thus, initial force values were set to zero. Finally, recordings were slightly shorter than 1,500 samples in some trials for technical reasons, so we examined the minimum number of samples in each participant, which ranged from 1,365 to 1,418 (M = 1,402, SD = 12.56). This slight deviation in recording length affected only the end, but not the beginning of the force trace. For plotting the group-average force trace in S2A Fig, we therefore used 1,365 samples (corresponding to 2,730 ms), and for individual force traces in Figs 2A–2C, S1 and S2B, we cropped all force traces to each participant's minimum sample size. Note, however, that importantly, these deviations in force trace recording length did not affect any of the measurements that went into our main results, which we describe in the following.

To investigate effort learning, we identified three measures at different stages of the force traces.

*Force initiation.* We defined the "start RT" (Fig 2A) as a measure of the time of force initiation. It was defined as the time that elapsed between the display of the thermometer and the participant's squeeze onset. To determine the squeeze onset for each trial, (1) we subtracted the average force of the first 10 samples from the smoothed force trace (smoothing kernel: 40 samples or 80 ms at 500 Hz) to make sure the initial force was zero; (2) we then computed the derivative of the smoothed force trace; (3) finally, we identified the earliest time point in the force trace where both the smoothed force exceeded 0.01 and its derivative surpassed a threshold of 0.001. In 4.6% of all trials where the maximum derivative occurred later than 400 samples, we adjusted the derivative threshold to 0.0001. All trials were subsequently visually inspected blind to condition, and the start onset was manually corrected if necessary. This only affected 1% of trials. Finally, for each participant, outlier trials where the start RT was more than three SD away from the mean of that participant were excluded, leaving more than 97% of all trials for each participant for further analysis.

*Effort priors.* The force at the first plateau in the force trace was used to represent the effort prior (Fig 2B). To define the first plateau, we looked for the first time point with a small rate of change in the force trace that occurred after the start RT. To determine this time, (1) we computed the derivative of the raw force trace; (2) we smoothed the derivative of the force trace (smoothing kernel: 40 samples); (3) we computed the absolute of the smoothed derivative; (4) and finally, we identified the smallest rate of change in the force trace as the first time point after the start RT where the force exceeded 0.15 MVC and where the absolute smoothed derivative was smaller than 0.001. Again, all trials were inspected to verify the definition of the effort prior to visually. Finally, outlier trials where the time of the prior was more than three SD away from the mean prior time of that participant were excluded, leaving 94.53%–97.92% of trials across participants (M = 96.74%, SD = 0.79%) for further analysis.

*Force adjustment.* The time that elapsed between reaching the effort prior and reaching a steady force at the required level was used to represent the adjustment RT (Fig 2C). To determine the onset of the steady state of the produced force, we implemented a few steps to ensure robustness because force traces naturally vary even when participants try to hold grippers at a stable force level. To define this onset as robustly as possible, (1) we computed the absolute smoothed derivatives of the force trace as done above; (2) to quantify local variability in force at a given point in time, we summed the absolute smoothed derivative in a sliding window of 300 samples (name 'summed derivatives' for short); (3) to detect the smallest change of local variability (i.e., when the force trace stabilised), we computed the absolute smoothed derivative of the summed derivative (name 'derivative of summed derivative' for short); (4) we identified the earliest time point after the effort prior at which three criteria were simultaneously satisfied: (a) the derivative of summed derivative was smaller than 0.0001, (b) the summed derivative was smaller than 0.1 (to exclude trials with high local variability and frequent fluctuations), and (c) the force exceeded 0.1. In later stages of the trial, the force trace sometimes became bumpy, which made it difficult to identify the steady state even by eye. The above procedure allowed us to detect the onset of the steady state for 80.47%–97.40% of trials across all participants (M = 92.05%, SD = 4.34%).

Following careful inspection of the data, seven participants were excluded due to poor task engagement based on fulfilling one or more of the following criteria: (1) the required effort was reached and maintained for at least 1,000 ms in less than 80% of trials (four participants; mean duration of force within effort range was 1,600 ± 233 ms across all 35 participants); (2) failure to follow the task instruction to adjust effort to required levels (e.g., pressed at a constant force close to their maximum on all trials; two participants); (3) unusually variable and noisy effort traces such that the percentage of trials in which a steady state was detected was lower than 80% (three participants).

**Regression models.** To determine the task factors that influence force initiation, we used MATLAB's function 'glmfit' to perform a linear regression with start RT as the dependent variable. We included seven regressors as independent variables, encoding the effects of volatility, noise, required effort in the previous trial and their interactions:

$$start\ RT \sim \beta_0 + \beta_1 \cdot vol + \beta_2 \cdot noise + \beta_3 \cdot effort_{t-1} + \beta_4 \cdot vol \cdot noise$$
$$+ \beta_5 \cdot vol \cdot effort_{t-1} + \beta_6 \cdot noise \cdot effort_{t-1} + \beta_7 \cdot vol \cdot noise \cdot effort_{t-1}$$

where *vol* is a binary vector encoding block-wise low (−1) or high (1) volatility levels, and *noise* is a binary vector encoding block-wise low (−1) or high (1) noise levels. All the regressors were zscored for fitting the regression. Beta coefficients were estimated for each regressor in each participant. Statistical inference was made at the group-level by testing beta coefficients across participants against zero using two-tailed t-tests.

To further test whether the effects of noise on start RT were confounded by long-term fatigue, we fitted a linear regression (MATLAB function fitlm) to the start RT as a function of trial number in each set of two adjacent blocks with the same hand. The estimated coefficients were used as the slope that reflects the rate of increase in the start RTs. Then, a repeated-measures ANOVA was performed on the slope of the start RT across the four pairs of blocks (block1 + 2, block3 + 4, block5 + 6 and block7 + 8).

To determine the time horizon (i.e., number of previous trials) that influence effort expectations, we applied a linear regression model with the effort priors as the dependent variable. We included three regressors as independent variables, encoding the effects of required effort in the past three trials:

$$effort\ prior \sim \beta_0 + \beta_1 \cdot effort_{t-3} + \beta_2 \cdot effort_{t-2} + \beta_3 \cdot effort_{t-1}$$

The change-point trials were excluded from the analysis. To test whether past effort could predict current priors, we performed one-sample t-tests against zero on the obtained beta weights (because here, any influence would only be of interest if it was positive, justifying a one-sided test). To investigate the influence of environmental statistics on effort priors, we fitted separate regression models for low and high volatility and low and high noise blocks, looking at the influence of the preceding three trials (trial *t-3*, trial *t-2,* and trial *t-1*). The obtained beta weights were compared in a repeated-measures ANOVA with factors noise, volatility, and trial.

To determine the influence of environmental statistics on prior adjustment, we applied a linear regression model with the adjustment RT as the dependent variable. We included five regressors as independent variables, encoding the effects of volatility, noise, and jump from the previous trial to the current trial:

$$adjustment\ RT \sim \beta_0 + \beta_1 \cdot vol + \beta_2 \cdot noise + \beta_3 \cdot jump\ sign + \beta_4 \cdot absolute\ jump\ size + \beta_5 \cdot vol \cdot noise$$

where *vol* and *noise* are two binary vectors encoding block-wise low (−1) or high (1) levels, and *jump sign* is a binary vector encoding jump down from high to low effort level (−1) or jump up from low to high effort level (1). Absolute jump size is the absolute difference in required effort between the current and previous trial, where larger values indicate the change in required effort is larger. Finally, a two-tailed *t* test (against zero) was applied to the beta values for each regressor's coefficient in each participant to make statistical inference at the group-level. To demonstrate the robustness of the obtained

volatility effect, regression models explaining adjustment RTs were compared when fitted with all trials or without change trials.

**Reinforcement-learning model.** Given that the effort priors are formed from previous experience, we fitted reinforcement-learning models to the subjects' priors. The model used a simple delta-learning rule to estimate the next prior given the past prior and the prediction error. The effort priors are updated by using the following equations:

$$\delta_t = E_t - P_t$$

$$\mu_{t+1} = P_{t+1} - P_t$$

where $E_t$ is the required effort for the current trial, $P_t$ is the effort prior for the current trial, $\delta_t$ is the prediction error at the current trial. Also, $\mu_{t+1}$ is the prior updating for the next trial, which is the prior difference between the next trial and the current trial.

To estimate average learning rates for different volatility and noise environments, we applied linear regression models separately to the four block types. We included prior updating as the dependent variable, and prediction error as the independent variable:

$$\mu_{t+1} \sim \beta_0 + \alpha \cdot \delta_t$$

In this way, a learning rate coefficient is gained for each condition. Finally, a two-way repeated-measures ANOVA was applied to the learning rate coefficients to determine the effects of volatility and noise on learning.

## Supporting information

**S1 Data. Excel spreadsheet containing individual participant data for** Figs 3A, 3B, 4C, 5A, 5B, **S3**, S4A, S4B, S5B, S5C, and S6A–S6D.
(XLSX)

**S1 Text. Supplementary results.** A full description of the results for all supplementary figures.
(DOCX)

**S1 Fig. Raw force traces in representative participants.** Representative participants in the categories of participants with the best **(A)**, average **(B)** and worst (but not excluded) **(C)** overall trace quality are shown with two consecutive blocks (96 trials) in low and high noise blocks, respectively.
(TIFF)

**S2 Fig. Force traces show evidence of effort learning.** Both average force traces **(A)** and examples from randomly selected participants **(B)** showed consistent patterns: effort priors at the initial plateau overshot the required effort in change down trials and undershot the required effort in change up trials, with both errors reduced in the trial that followed the change trial (trial $t+1$). **In both (A) and (B),** blue colour indicates the change trial $t$, and purple indicates the following trial $t+1$. The solid line indicates the force trace. The dashed line indicates the required effort. The star indicates the effort prior. (A) The error bars in force traces indicate the standard error across participants at each time point. The vertical error bars indicate the standard error of the time point of the effort prior. (B) The black dashed line indicates the required effort in trial $t-1$, the blue dashed line indicates the required effort in trial $t$, and the purple dashed line indicates the required effort in trial $t+1$.
(TIFF)

**S3 Fig. The estimated prior is not driven by sensorimotor feedback.** The difference between the current effort prior and current required effort is plotted for change trials with different jump sizes across all participants. The jump size is defined as the change in required effort from trial *t* to trial *t-1* independent of noise. Given the jump could not be anticipated, if the prior merely reflects an expectation formed on previous trials, rather than sensorimotor feedback accumulated on the current trial, the error should perfectly scale with the jump size. This was indeed what was observed: the prior's deviation from the required effort scaled with increased jump size. The observed slope (red line) was close to the ideal slope (slope = −1; black line), and deviations were smaller for upwards change trials. This shows that any influences of immediate sensorimotor feedback on our estimate of prior were minimal. All data used to generate this plot can be found in S1 Data.
(TIFF)

**S4 Fig. No evidence of long-term fatigue effects on force initiation. (A)** Group means estimates derived from a linear regression show the effect of block-wise noise difference, trial number and last effort on start RT. Start RT did not differ between the first two block groups (block 1 + 2 versus block 3 + 4; all low noise blocks) or the last two block groups (block 5 + 6 versus block 7 + 8; all high noise blocks). Thus, the increase of start RTs over blocks is unlikely to be driven by long-term fatigue. Stars indicate significance from two-tailed one-sample t-tests on the obtained parameter estimates. **(B)** Slopes were fitted to start RTs over trials (trial 1–96) within each block group and slope estimates are shown here across block groups. This confirmed that the slowing of start RTs with progression through a block group was similar across all block groups of the task. The rate of increase in start RTs did not increase with time across block groups, supporting the interpretation that noise effects observed in Fig 3 are unlikely to be confounded by long-term fatigue. The error bar indicates the standard error. $^{***}$ $P < 0.001$, $^{ns}$ $P > 0.05$. All data used to generate this plot can be found in S1 Data.
(TIFF)

**S5 Fig. Effort learning is not influenced by motor noise. (A)** An example force trace illustrates the definition of onset (green circle) and offset (red square) of the stable period (grey shading). The standard deviation of the force trace within this stable period was extracted as a measure of motor noise for each trial. **(B)** To validate our measure of motor noise before examining relationships with learning, we tested whether motor noise scales with required effort. The significant slope (red line) shows that motor noise increased with higher effort. **(C)** Group means estimates from a regression show the effects of the previous trial's required effort and motor noise on absolute prior updating. There was no evidence to suggest that motor noise contributed to trial-by-trial learning. **(D)** Subject-wise correlation between motor noise and learning rate coefficients from the reinforcement-learning model. Again, there was no evidence for a relationship between motor noise and learning across participants. All data used to generate this plot can be found in S1 Data.
(TIFF)

**S6 Fig. Control analyses of within-trial force adjustments (adjustment RTs). (A)** Group means estimates from a regression equivalent to that reported in Fig 5B, this time after exclusion of change trials. Our key results replicated here: adjustment RTs were faster under higher noise but slower under higher volatility environments. Also, trial-by-trial changes in effort—the direction and absolute magnitude of jump both slowed down adjustment RTs. The only difference from the main Fig 5A is that the current regression showed a relatively weaker interaction (i.e., smaller *t* value and larger *p* value) between volatility and noise environments. **(B)** Adjustment RTs in high and low volatility environments, across low and high noise environments, after exclusion of change trials. This replicates the interaction between noise and volatility as in Fig 5B: in low noise environments, adjustment RTs were slower in high volatility compared to low volatility; in high volatility environments, adjustment RTs were faster in high noise compared to low noise environments. The only difference from the main Fig 5B is that the current interaction showed relatively weaker (i.e., smaller *t* value and larger *p* value) volatility effect (i.e., slower adjustments under higher volatility) in low noise environments. **(C)** To test whether stability after mean effort switches leads to faster adjustment RTs, we compared adjustment RTs between early and late stages within a

constant mean effort level. To make different stages within each constant mean effort level comparable in different volatility blocks, we considered the first five trials as an early stage in low volatility, and the later five trials as a late stage in low volatility, as well as the first five trials as an entire phase in high volatility (equivalent to the early stage of a low volatility block). This is because the required effort changes every 10–14 trials in low volatility and the required effort changes every 5–7 trials, which making the entire phase in a high volatility block correspond to the early stage of a low volatility block, but no trials in a high volatility block could correspond to the late stage of a low volatility block. Thus, here we averaged adjustment RTs separately for the early and late stages of a constant mean effort period in low volatility blocks but compared it to the entire constant mean effort period of high volatility blocks, across low and high noise blocks. This shows an interaction between noise and volatility, with faster adjustments in the late stage of a constant mean effort low volatility phase, compared to both high volatility and late stages of a constant mean effort low volatility phase. **(D)** Same analysis and results as in (C), but with change trials excluded. The error bar indicates the standard error. * $P < 0.05$, ** $P < 0.01$, *** $P < 0.001$. All data used to generate this plot can be found in S1 Data.
(TIFF)

## Acknowledgments

The views expressed are those of the author(s) and not necessarily those of the NIHR or the Department of Health and Social Care.

## Author contributions

**Conceptualization:** Rong Bi, Miriam C. Klein-Flügge, Lilian Weber.

**Data curation:** Rong Bi.

**Formal analysis:** Rong Bi, Jan Grohn.

**Funding acquisition:** Miriam C. Klein-Flügge.

**Investigation:** Rong Bi.

**Methodology:** Rong Bi, Jan Grohn, Miriam C. Klein-Flügge, Lilian Weber.

**Project administration:** Miriam C. Klein-Flügge.

**Software:** Patricia L. Lockwood, Miriam C. Klein-Flügge.

**Supervision:** Miriam C. Klein-Flügge, Lilian Weber.

**Visualization:** Rong Bi.

**Writing – original draft:** Rong Bi.

**Writing – review & editing:** Rong Bi, Jan Grohn, Patricia L. Lockwood, Miriam C. Klein-Flügge, Lilian Weber.

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
