## [Editor Report · Decision Letter 0]

30 Oct 2025

Dear Dr Bi,

Thank you for submitting your manuscript entitled "Environmental Uncertainty Shapes Human Effort Learning" for consideration as a Research Article by PLOS Biology.

Your manuscript has now been evaluated by the PLOS Biology editorial staff as well as by an academic editor with relevant expertise and I am writing to let you know that we would like to send your submission out for external peer review.

Once your full submission is complete, your paper will undergo a series of checks in preparation for peer review. After your manuscript has passed the checks it will be sent out for review. To provide the metadata for your submission, please Login to Editorial Manager (https://www.editorialmanager.com/pbiology) within two working days, i.e. by Nov 03 2025 11:59PM.

Kind regards,

Luke

(on behalf of my collage Christian, who is the handling editor for your paper, but who is off this week)

Lucas Smith, Ph.D.

Senior Editor

PLOS Biology

lsmith@plos.org

--on behalf of--

Christian Schnell, PhD

Senior Editor

PLOS Biology

---

## [Decision Letter · Decision Letter 1]

18 Dec 2025

Dear Dr Bi,

Thank you for your patience while your manuscript "Environmental Uncertainty Shapes Human Effort Learning" was peer-reviewed at PLOS Biology. It has now been evaluated by the PLOS Biology editors, an Academic Editor with relevant expertise, and by several independent reviewers.

In light of the reviews, which you will find at the end of this email, we would like to invite you to revise the work to thoroughly address the reviewers' reports.

You will see that the reviewers think that the study is overall well executed and provides important insights. However, they all request clarifications and a few additional analyses which we ask you to address in a revision.

Given the extent of revision needed, we cannot make a decision about publication until we have seen the revised manuscript and your response to the reviewers' comments. Your revised manuscript is likely to be sent for further evaluation by all or a subset of the reviewers.

**IMPORTANT - SUBMITTING YOUR REVISION**

*Re-submission Checklist*

*Published Peer Review*

*PLOS Data Policy*

*Blot and Gel Data Policy*

Sincerely,

Christian

Christian Schnell, PhD

Senior Editor

PLOS Biology

cschnell@plos.org

REVIEWS:

Reviewer #1 (Nils B. Kroemer, signed his report): In their submission "Environmental Uncertainty Shapes Human Effort Learning", Bi et al. present an innovative approach to estimate the effects of uncertainty on human effort learning using a novel task design. In my opinion, this is already a strong and convincing contribution, and I have only minor suggestions to potentially further improve the submission. Strengths of the paper include the use of a sound and elegant task design, the clear description of the rationale and the methods, the high quality of the data visualization, and the overall quality of the writing and reporting of the findings. I think this is one of the best papers I have reviewed this year, and I would like to commend the authors for this relevant contribution to the effort learning literature.

Minor issues

1. The differences in learning-related adjustments between noise and volatility are very interesting, including the difference to reward learning. However, unless I missed it, I did not see whether the authors analyzed correlations between the estimated adjustments to noise or volatility at the level of the individuals. In other words, is someone who is more likely to learn from noise also more likely to learn from volatility (even if the directions are opposite) or are those independent aspects of effort learning? Likewise, I was wondering how independent the different estimates of the learning process are (e.g., force initiation or adjustment RT). These additional analyses may strengthen the interpretation of the findings.

2. Although the differences between the pattern of effort vs. reward learning adjustments to noise are intriguing, I would be cautious to interpret this more strongly based on the evidence provided here. First, the same participants did not complete a reward learning task. Second, the effort learning task did not manipulate reward levels, which might have helped to further elucidate the link to reward-related computations. I would suggest stating more clearly that the study cannot fully answer this question and that future research should follow-up on this potential discrepancy.

3. The number of excluded participants is quite high, although the stated criteria seem reasonable to me. Please specify if the exclusion criteria were defined a priori and how generalizable the findings are in this sample. Are there participants where the model does not provide a good fit of the behavioral data?

4. Relatedly, were any alternative models explored (and compared to the final model)?

5. Could the authors comment on the subjective experience of participants? I agree that the conceptual link to energy metabolism is of interest, but a related question is whether the participants are consciously aware of their behavioral adjustments to noise and volatility. Did the authors ask any questions about the strategy of the participants and/or their subjective level of exertion?

6. In Figure 1, the caption for the high noise trials is misspelled (as how). Moreover, I found the visualization of either high or low noise trials a bit difficult to understand initially because I thought that they alternated from trial to trial. Maybe this can be improved.

Signed,

Nils B. Kroemer

Reviewer #2: The manuscript by Bi and colleagues investigates how humans learn required motor effort during decision making. Most studies investigating effort-based decision making explicitly indicate the required effort to obtain a reward to participants at the outset of each trial. Here, the authors investigate how this effort is learnt from trial and error, and how this learning is shaped by environmental statistics. They focus on two properties of the environment, noise (irreducible or expected uncertainty) and volatility (or unexpected uncertainty), that have been frequently investigated in the reward-based domain. The authors focus on three stages and how they are influenced by task parameters: force initiation, prior (reflecting participants' estimate of the required effort) and adjustment in response to sensory feedback. Overall, I think this is an interesting study that is carefully designed and analyzed, the analyses appear straightforward and well described, and the data support the authors' conclusions. The manuscript is well written and easy to read. I actually only have a few rather minor comments:

1. The definition of the "prior" appeared somewhat surprising to me. By the authors' reasoning, this is thought to reflect participants' estimate of how much effort will be required on the current trial. Here, the prior is defined as the point at which, after force initiation, the force trace plateaus. I would argue that, at this stage, people already integrate the sensorimotor feedback they receive online with their prior estimate of required effort. I would not be too adamant about this point, as the data clearly indicate sensitivity of the prior across trials (Fig. 2F), but this does not preclude that this measure also included variance relate to online sensorimotor control

2. In the abstract, it is mentioned that effort learning is investigated "across different stages of the effort production process (e.g., initiation, prior, adjustment)" - I guess, at this stage, the reader probably cannot have a detailed idea what these phases are - in particular, "prior" to me sounded like a statistical quantity (a prior estimate/knowledge/probability), not a temporal stage of a trial. Perhaps either omit - or add a short sentence describing this?

3. The regression results in 4B show an effect of past effort predominantly from trials t-1 and t-2 on the current prior, with a clear effect of noise, but not volatility. This is strongest on t-1. This finding seems to contrast with the learning rate coefficients in 4C, which are affected by both noise and volatility. What is the reason for this discrepancy?

4. Regression models investigating start/adjustment RT: do these RT measures also show the typical non-Gaussian (skewed) distribution? If so, was log(RT) used for the analyses?

5. Maybe I've missed this, but how were people incentivized/rewarded?

Reviewer #3: This study investigates how healthy individuals learn to regulate effort in response to varying environmental uncertainties. Using a novel effort-learning task with a handheld dynamometer, the experimental task manipulates two forms of uncertainty: volatility by varying the number of exertion levels and noise by altering feedback from the effort-exertion cursor. The findings show that participants effectively estimate required effort by integrating prior experiences and immediate sensorimotor feedback. Importantly, the study reveals that environmental conditions significantly influence effort-learning dynamics, with high-noise environments resulting in slower initiation, weaker effort priors, and rapid within-trial adjustments; while high-volatility contexts lead to slower learning and adjustments. The results suggest that different sources of uncertainty are incorporated into internal effort estimates, providing insights into the mechanisms of effort learning. This study explores a timely question — effort learning — which has received limited investigation. The experiments are well designed and the analytical approaches used are suitable. I have some suggested analyses to strengthen the manuscript.

The authors manipulate noise by providing visual feedback of exertion sampled from different distributions. It appears that the actual force traces illustrated in Figure S1 are quite noisy — resulting from motor variability/noise. It would be important to show that the actual motor noise on a trial is not related to learning rates. This motor noise could be another form of uncertainty that impacts learning (in addition to the volatility and 'visual' noise manipulated in the experiment). The authors should perform analyses to account for the motor variability/noise factor.

Another issue that should be considered is the idea of signal-dependent noise — exertion variability increases with increasing exertion level (Jones et al., 2002; Haris and Wolpert, 1998). While the authors control visual feedback in the experiment's noise condition, there is still motor noise that scales with exertion levels. It would be important to control for the motor noise to evaluate how this form of uncertainty might contribute to learning, keeping in mind that it will scale with the levels of exertion.

Related to this signal-dependent noise idea, several studies of effort-based judgements have shown that variability in exertion is related to increased feelings of effort (Salimpour and Shadmehr, JNeuro, 2014; Salimpour, Mari, Shadmehr, JNeuro, 2015; Hu et al, JNeuro, 2022; Padmanabhan et al, npj Parkinsons, 2023;), and the authors should discuss their findings in the context of this previous work.

---

## [Editor Report · Decision Letter 2]

10 Apr 2026

Dear Dr Bi,

Thank you for your patience while we considered your revised manuscript "Environmental Uncertainty Shapes Human Effort Learning" for publication as a Research Article at PLOS Biology. This revised version of your manuscript has been evaluated by the PLOS Biology editors and the Academic Editor.

Based on our Academic Editor's assessment of your revision, we are likely to accept this manuscript for publication, provided you satisfactorily address the following data and other policy-related requests:

* Please add the links to the funding agencies in the Financial Disclosure statement in the manuscript details.

* Please include information in the Methods section whether the study has been conducted according to the principles expressed in the Declaration of Helsinki.

* DATA POLICY:

Regardless of the method selected, please ensure that you provide the individual numerical values that underlie the summary data displayed in the following figure panels as they are essential for readers to assess your analysis and to reproduce it: 3AB, 4 (bottom right panel), 5AB, S3, S4AB, S5BC and S6ABCD.

* CODE POLICY

Per journal policy, if you have generated any custom code during the course of this investigation, please make it available without restrictions. Please ensure that the code is sufficiently well documented and reusable, and that your Data Statement in the Editorial Manager submission system accurately describes where your code can be found. More information on our Code Policy, what and how to share can be found here: https://journals.plos.org/plosbiology/s/code-availability

We expect to receive your revised manuscript within two weeks.

*Published Peer Review History*

*Press*

Sincerely,

Christian

Christian Schnell, PhD

Senior Editor

cschnell@plos.org

PLOS Biology

---

## [Editor Report · Decision Letter 3]

24 Apr 2026

Dear Rong,

Thank you for the submission of your revised Research Article "Environmental Uncertainty Shapes Human Effort Learning" for publication in PLOS Biology. On behalf of my colleagues and the Academic Editor, Thorsten Kahnt, I am pleased to say that we can in principle accept your manuscript for publication, provided you address any remaining formatting and reporting issues. These will be detailed in an email you should receive within 2-3 business days from our colleagues in the journal operations team; no action is required from you until then. Please note that we will not be able to formally accept your manuscript and schedule it for publication until you have completed any requested changes.

While you attend to these requests, can you please also add a pointer to the source data to the corresponding figure legends of the supplementary information (in the same way as you have done already for the main figures)?

PRESS

We frequently collaborate with press offices. If your institution or institutions have a press office, please notify them about your upcoming paper at this point, to enable them to help maximize its impact. If the press office is planning to promote your findings, we would be grateful if they could coordinate with biologypress@plos.org. If you have previously opted in to the early version process, we ask that you notify us immediately of any press plans so that we may opt out on your behalf.

Sincerely,

Christian

Christian Schnell, PhD

Senior Editor

PLOS Biology

cschnell@plos.org